# A Quantitative Investigation of Mental Fatigue Elicited during Motor Imagery Practice: Selective Effects on Maximal Force Performance and Imagery Ability

**DOI:** 10.3390/brainsci13070996

**Published:** 2023-06-26

**Authors:** Franck Di Rienzo, Vianney Rozand, Marie Le Noac’h, Aymeric Guillot

**Affiliations:** 1Univ Lyon, Université Claude Bernard Lyon 1, Laboratoire Interuniversitaire de Biologie de la Motricité, EA 7424 Villeurbanne, France; franck.di-rienzo@univ-lyon1.fr (F.D.R.); marie.lenoach97@gmail.com (M.L.N.); 2Université Jean Monnet Saint-Etienne, Lyon 1, Université Savoie Mont-Blanc, Laboratoire Interuniversitaire de Biologie de la Motricité, F-42023 Saint-Etienne, France; vianney.rozand@univ-st-etienne.fr

**Keywords:** mental practice, mental fatigue, motor performance, recuperation

## Abstract

In the present study, we examined the development of mental fatigue during the kinesthetic motor imagery (MI) of isometric force contractions performed with the dominant upper limb. Participants (*n* = 24) underwent four blocks of 20 MI trials of isometric contractions at 20% of the maximal voluntary contraction threshold (20% MVC_MI_) and 20 MI trials of maximal isometric contractions (100% MVC_MI_). Mental fatigue was assessed after each block using a visual analogue scale (VAS). We assessed maximal isometric force before, during and after MI sessions. We also assessed MI ability from self-report ratings and skin conductance recordings. Results showed a logarithmic pattern of increase in mental fatigue over the course of MI, which was superior during 100% MVC_MI_. Unexpectedly, maximal force improved during 100% MVC_MI_ between the 1st and 2nd evaluations but remained unchanged during 20% MVC_MI_. MI ease and vividness improved during 100% MVC_MI_, with a positive association between phasic skin conductance and VAS mental fatigue scores. Conversely, subjective measures revealed decreased MI ability during 20% MVC_MI_. Mental fatigue did not hamper the priming effects of MI on maximal force performance, nor MI’s ability for tasks involving high physical demands. By contrast, mental fatigue impaired MI vividness and elicited boredom effects in the case of motor tasks with low physical demands.

## 1. Introduction

Motor Imagery (MI, the mental representation of an action without engaging in its physical execution) is one of the most prevalent mental techniques in sport psychology to improve performance [1,2]. MI can impact a variety of factors mediating athletic performance, such as technical execution of skills, physiological arousal but also motivation and confidence [3]. From a metabolic standpoint, MI represents an energy-intensive cognitive process [4]. This is attested by functional brain imaging reports of BOLD-signal increases within cortical and subcortical brain structures overlapping those mediating the actual motor preparation and execution of the same task [5,6]. The repetitive activation of these brain structures through MI can induce mental fatigue, defined as a psychobiological state of decreased cognitive performance resulting from prolonged or excessive cognitive activity [7,8]. In spite of a general consensus that MI elicits mental fatigue and practice guidelines advocating MI sessions of limited durations—i.e., below the ~20 min threshold—to prevent its onset [9,10,11], the relationship between MI and mental fatigue has been the focus of a very limited number of experimentations.

Rozand et al. [12] were the first to address whether mental fatigue induced by MI practice might negatively impact physical performance. The authors found that mental fatigue elicited by 80 consecutive MI trials did not impair the maximal isometric force of the elbow flexor muscles. Likewise, they observed no deleterious effects of mental fatigue elicited by MI practice on the central activation ratio obtained from transcranial magnetic stimulation. Such absence of negative effects was later confirmed by the same group, although mental fatigue was elicited using a Stroop task and not through MI practice [13]. Importantly, they focused on maximal isometric force peaks during 5 s voluntary contractions and used a single maximal isometric force trial to assess force performance. Possibly, a different results pattern could have emerged from measures of the total force output over longer contraction periods or over the course of multiple trials. For instance, recent electroencephalography data indicated increased motor-related cortical potential and muscle activation to achieve submaximal isometric contractions of knee extensor muscles after the induction of mental fatigue through 200 MI trials [14]. Also, Nakashima et al. [15] measured the effect of mental fatigue induced by physical or mental fatigue on a visuomotor control task requiring submaximal force levels. Mental fatigue was assessed before and after 200 repetitions of the task, using subjective self-reports and transcranial magnetic stimulation measures. The authors observed a parallel pattern of mental fatigue development and alteration of motor performances under the two practice conditions.

Overall, mental fatigue induced by MI practice appeared to have negative effects on endurance paradigms involving submaximal contractions or speed-accuracy tradeoffs [14,15,16]. By contrast, maximal isometric force performance was shown unaffected [12]. Nonetheless, quantitative insights into the development of mental fatigue over the course of a MI practice session are lacking. Indeed, mental fatigue is usually assessed before and after, but not during MI practice sessions. Furthermore, whether mental fatigue elicited by MI practice affects MI ability remains currently unknown. This issue is of critical importance due to the deleterious effects of mental fatigue on cognitive–motor processes [17]. MI ability can be assessed from psychometric, behavioral and neurophysiological methods [18,19,20]. Thorough assessment should ideally combine both subjective and objective measures ([20] for an illustration in clinical populations). There is accumulating evidence supporting that the benefits of MI practice on motor performance are related to MI ability [21,22,23,24]. Better imagers usually achieve greater behavioral improvements than poor imagers and elicit higher levels of brain motor networks stimulation during the mental rehearsal of movements [25,26]. It is noteworthy, reciprocally, that the level of motor expertise predicts a higher MI ability as well as focused and intense activation of brain motor regions during MI [27]. Conversely, poor imagers exhibited diffuse activation patterns, thus reducing the leverage of MI practice on experience-based plasticity within cortical and subcortical regions of brain motor systems [28]. Overall, mental fatigue could hamper the efficacy of MI practice interventions through reduced MI ability. For instance, recent data indicate that mental fatigue interfered with the classification of electroencephalographic signals, which thus might restrict MI use in brain–computer interfaces [29]. A final neglected variable of interest is the content of MI in experimental paradigms designed to investigate its effects on mental fatigue. Due to the tight correspondence in brain activation patterns during MI and physical practice of the same task, up to reproduce its physical demands [30,31], one might assume that MI of intense physical efforts could yield additional mental fatigue and have greater effects on maximal force production capacity.

The primary aim of this study was to investigate the development of mental fatigue during a prolonged MI practice session and establish whether mental fatigue influenced MI ability. Spurred by the pioneering work by Rozand et al. [12], we hypothesized that the decline threshold due to the onset of mental fatigue would occur between 40–80 trials of MI. The second aim was to address whether repeated MI practice of submaximal vs. maximal contractions yielded comparable mental fatigue and their effects on maximal force production capacity. We hypothesized an earlier onset and greater amount of mental fatigue under MI practice of maximal contractions compared to submaximal contractions associated with a greater impairment in maximal force.

## 2. Materials and Methods

### 2.1. Participants

Twenty-four healthy adults volunteered to participate in the study. Participants had no medical history that could have compromised the results including functional limitations of the upper limb. They were off medication and instructed to not consume alcohol or modify their caffeinated beverage habits the days of experimentation. We included participants aged 20–30 years old (body mass index ranging 19–24 kg·m^−2^), with a regular practice of physical or sporting activities (>2 sessions of 1 h/week over the last 6 months). We also screened for low MI ability by including participants with a score > 5 out of 7 on kinesthetic subscale of the French translation of the MIQ-3f questionnaire, which assesses MI ease on a Likert-type scale ranging from 1 = “Very hard to perceive” to 7 = “Very easy to perceive” [32]. The experiment received ethical approval by the local review board of the University. Participants provided a written informed consent form before their enrolment in the study, in agreement with the principles and statements laid out in the Declaration of Helsinki (1964) and its later amendments. They received no information regarding the purpose of the experiment until after completion of the design.

### 2.2. Experimental Design

We implemented a repeated measures experimental design involving two experimental sessions separated by 48–96 h (Figure 1). Experimental sessions consisted in (i) baseline evaluations, immediately followed by (ii) a session of MI practice.

#### 2.2.1. Baseline Evaluations

We first assessed participants’ mood from the BRUMS questionnaire [33]. The BRUMS questionnaire contains 24 items, divided into six subscales (i.e., anger, confusion, depression, fatigue, tension and vigor). Participants provide ratings on a 5-point Likert-type scale ranging from 0 = “not at all” to 4 = “extremely”. Scores, thus, range 0–16 for each subscale. Since we aimed at controlling the fatigue state at baseline, we only considered the fatigue subscale of the questionnaire (i.e., “exhausted”, “tired”, “sleepy” and “worn-out” items), with a specific consideration for the “sleepy” item due to established harmful effects of sleep deprivation on cognition [34]. We then assessed maximal force performance of the elbow flexor muscles. Participants sat on an adjustable reclining seatback in front of an inverted force plate, with their elbow at 90° in contact with their trunk (see [35,36] for a similar procedure). They adopted a standardized position, keeping the back of their head, trunk and sacrum in contact with the reclining seatback. A standardized warm-up consisted of 6 incremental isometric contractions, sustained for 10 s and separated by 10 s of rest. They were instructed to start from ~20% of their maximal voluntary contraction (MVC) and reach 90% of the MVC threshold on the 6th trial. Then, participants performed 2 MVC, separated by a period of passive recovery of 1.5 min. Only 5% of difference was tolerated between the 2 trials to be considered representative of the pretest MVC threshold. Participants ended the motor performance evaluation by 4 additional isometric contractions at 20% of the MVC sustained for 10 s, similarly to what they would have to imagine during the experimental condition. After motor performance evaluations, participants imagined the same voluntary isometric contractions paradigm using kinesthetic MI, i.e., 6 incremental contractions, 2 MVC and 6 contractions at 20% of the MVC. This aimed at getting participants familiar with MI before engaging in the MI practice session. We then collected the perceived workload post-warm up from the NASA-TLX [37]. The NASA-TLX measures 6 different dimensions (“mental demand”, “physical demand”, “temporal demand”, “performance”, “effort” and “frustration”) on a 21-point Likert-type scale. For the present study, we only analyzed results from the mental demand, physical demand, temporal demand and effort subscales as indicators of the workload of the MI practice session. We also indexed the state of mental fatigue using a visual analogous scale of 10 cm (VAS) ranging from 0 = “Complete absence of mental fatigue” to 10 = “Complete mental exhaustion”. Participants finally reported subjective scores of MI difficulty and vividness on a 10-point Likert-type scale ranging from 0 = “Absence of sensations/difficulty” to 10 = “Identical sensations as the physical practice of the force task/Very high difficulty”.

#### 2.2.2. Experimental Conditions

Immediately after the baseline evaluations, participants completed a MI practice session expected to elicit mental fatigue. The session consisted of 4 blocks of 20 kinesthetic MI trials of either maximal or submaximal isometric contractions. We chose to administer kinesthetic rather than visual MI since (i) it is generally considered a more demanding form of mental practice [38,39] and (ii) because isometric contractions do not produce observable movements that could facilitate the build-up of visual mental representations [40]. Blocks were separated by 30 s of rest. Inspired by the methods in Rozand et al. [12], each MI trial consisted of [0, 10] s of kinesthetic MI, followed by [0, 10] s of rest. The timing of the MI practice session was externally cued by auditory stimuli, using Presentation^®^ software (Version 23.0, Neurobehavioral Systems, Inc., Berkeley, CA, USA). Participants were instructed to mentally reproduce the sensations associated with physical practice of the force task at different intensities. During the first experimental condition, participants completed the MI practice session using kinesthetic MI of isometric contractions of elbow flexor muscles against the force plate at 20% of their MVC (20% MVC_MI_ condition) sustained for 10 s. During a second experimental condition, participants completed an identical MI practice session but focused on voluntary isometric contraction of elbow flexor muscles at 100% of the MVC (100% MVC_MI_ condition). To control for carryover effects, experimental sessions were administered in a counterbalanced order at the group level.

After each block, participants reported their level of mental fatigue on the VAS, as well as their perceived MI vividness and difficulty on a 10-point Likert-type scale (Figure 1). One actual MVC trial was administered between the 2nd and 3rd block of MI practice, and 2 actual MVC trials were performed after the 4th block. This enabled us to index force performance throughout the MI practice session (Figure 1). Only one trial was administered between the 2nd and 3rd block, which was intended to limit the amount of time spent between blocks in the absence of MI practice, thereby limiting potential recovery from mental fatigue induced by repeated MI practice at the session-level. After the MI practice session and force evaluations, participants completed the BRUMS and NASA-TLX questionnaires (Figure 1).

### 2.3. Data Processing and Extraction

#### 2.3.1. Force Plate Recordings

Elbow flexion force was measured from an immovable force plate (AMTI, model OR6-7-203 2000, Watertown, MA, USA). Signals were sampled at 1000 Hz. After performing a frequency and residual analysis of raw signals, data were smoothed with a zero-lag low-pass Butterworth filter (4th order), with a cut-off frequency of 20 Hz. Abrupt increments of the total force were detected for each maximal isometric force trial using a threshold function (Matlab^®^, 2021). We first extracted the peak force over the [0, 10] s, corresponding to each maximal isometric force trial. The total force was then obtained by integrating the force slope with respect to each [0, 10] s trial window (trapezoid rules). We finally normalized the peak force and the total force as a percentage of the mean values obtained during pre-test maximal isometric force trials.

#### 2.3.2. Neurophysiological Recordings

##### Electromyographic and Electrocardiographic Recordings

We recorded the surface electromyograms (EMG) of the main agonist and antagonist muscles of the voluntary elbow flexion against the force plate. After shaving and cleaning the skin with alcohol, we collected EMG activity from the biceps brachii, anterior deltoideus and triceps brachii using pairs of surface electrodes (1 cm EMG Triode, nickel-plated brass, inter-electrode distance 2 cm, Thought Technology, Montreal, QC, Canada). We followed the “Surface Electromyography for the Non-Invasive Assessment of Muscles” (SENIAM) guidelines for electrode positioning [41]. Electrodes’ locations were marked with a pen and photographed to ensure reproducible positioning from one experimental session to another. Raw EMG signals were recorded and synchronized by LabChart ProV8© (ADInstruments Pty Ltd., Bella Vista, Australia, 2014) from the TrignoTM Wireless© EMG system (2014, Delsys Incorporated). We also continuously monitored the heart rate (HR) using a finger pulse compatible with the acquisition system. We then rectified and smoothed the raw EMG signal with a 20–500 Hz pass-band filter (Butterworth 4th order). To index muscle activation, we applied a root mean square filter across the [0, 10] s time window corresponding to maximal isometric force trials (*EMG_RMS_*). We then calculated an activation ratio by expressing agonist *EMG_RMS_* (*EMG_RMS_* from the biceps brahcii and anterior deltoideus) as a function of antagonist *EMG_RMS_* (*EMG_RMS_* from the triceps brachii), as follows:$${EMG}_{RATIO}=\frac{\left(Biceps\;brachii\;{EMG}_{RMS}+Anterior\;deltoideus\;{EMG}_{RMS}\right)}{Triceps\;brachii\;{EMG}_{RMS}}$$

*EMG_RATIO_* measures were finally expressed as a percentage of *EMG_RATIO_* recorded during the pretest trials.

##### Skin Conductance Recordings

We continuously recorded electrodermal activity by measuring skin conductance using two 50 mm^2^ unpolarizable bipolar electrodes, placed on the second phalanx of the second and third digits of the non-dominant hand (MLT116f GSR Finger Electrodes, ADInstruments, Dunedin, New Zealand). Electrodermal activity reflects the activity of eccrine sweat glands, which are under the unique control of the sympathetic branch of the autonomic nervous system. Increased skin conductance thus attests to increased sympathetic activity and inversely. Skin conductance recordings represent objective markers of MI ability since variations mirror changes in vigilance states and anticipation of energy expenditure related to action preparation [42]. There is also experimental evidence for an association between the vividness of MI and the amplitude of the electrodermal response [42]. We quantified the amplitude of electrodermal response (*EDR_AMP_*) as the delta in skin conductance during each [0, 10] s time windows allocated to MI trials:$${EDR}_{AMP}=Maximum\;{SC}_{\left[0,10\right]s}-Minimum\;{SC}_{\left[0,10\right]s}$$

### 2.4. Statistical Analysis

#### 2.4.1. Power Considerations

We determined the sample size to be able to detect medium to low effect sizes (i.e., 5–10% of explained variation) for the CONDITION (100% MVC_MI_, 20% MVC_MI_) by BLOCK (logarithmic regressor, 1 to 5) interaction effect on VAS scores with a power of p_1-β_ = 0.80. We used the pwr package [43] implemented in R [44], and the ad hoc functions for linear mixed effects models. This analysis yielded a sample size of *n* = 21 participants, which we increased to *n* = 24 to guard against potential dropouts.

#### 2.4.2. Analysis of the Dependent Variables

We analyzed the dependent variables using a series of linear mixed effects analyses with by-subject random intercepts, using R [44] and the package nlme [45]. To analyze VAS scores of mental fatigue, we entered the fixed effects of BLOCK (logarithmic regressor, 1 to 5) and CONDITION (100% MVC_MI_, 20% MVC_MI_) with interaction term. We used a comparable model to analyze peak force and the total force during maximal isometric contractions, although we included TEST (Pretest, Block 2–3, Block 4) as repetition factor. Subjective scores of MI ease/difficulty and vividness were analyzed with the interaction between CONDITION and BLOCK as fixed effects, but we also entered VAS scores as numeric regressors. This enabled us to investigate predictive relationships between the state of mental fatigue and subjective measures of MI ability. HR and *EDR_AMP_* were analyzed using a similar random-coefficient regression model. For subjective scores obtained from the BRUMS and NASA-TLX, we included the additional main effect of DIMENSION in the regression model formula to account for the effect of the different subscales of the questionnaire. Visual inspection of residual plots did not reveal any obvious deviations from homoscedasticity or normality. As effect sizes, we reported partial coefficients of determination (η_P_^2^) using the effectsize package [46]. The statistical threshold was set up for a type 1 error rate of α = 5%. Main and interaction effects were investigated post hoc using general linear hypotheses testing of planned contrasts from the multcomp package [47]. Holm’s sequential corrections were applied to control the false discovery rate.

## 3. Results

### 3.1. BRUMS and NASA-TLX Scores

BRUMS scores on the fatigue dimension were affected by the TEST × CONDITION interaction (η_P_^2^ = 0.08, χ^2^(1) = 3.79, *p* = 0.05). Post-hoc contrasts revealed that the increase from the Pretest (0.78, 95% CI [0.59, 0.97]) to the Posttest (1.25, 95% CI [0.99, 1.51]) during 100% MVC_MI_ outperformed the Pretest (0.70, 95% CI [0.53, 0.86]) vs. Posttest (1.00, 95% CI [0.78, 1.21]) increase during 20% MVC_MI_ (*p* = 0.05). Noteworthy, very low levels of sleepiness were reported for the “sleepy” item of the BRUMS (0.59 ± 0.90). This low level of sleepiness remained constant across experimental sessions (20% MVC_MI_: 0.47, 95%CI [0.25, 0.70]; 100% MVC_MI_: 0.71, 95% CI [0.41, 1.01]). NASA-TLX ratings were only affected by the main effects of TEST (η_P_^2^ = 0.06, χ(1) = 23.24, *p* < 0.001) and DIMENSION (η_P_^2^ = 0.09, χ(3) = 3.60, *p* < 0.001). This was due to higher ratings during the Posttest (9.29, 95% CI [8.56, 9.94]) compared to the Pretest (7.28, 95% CI [6.62, 7.96]) (*p* < 0.001) and reduced ratings for the Temporal demand (7.60, 95% CI [6.53, 8.68]) and Physical demand (6.70, 95% CI [5.80, 7.60]) compared to the Mental demand (9.67, 95% CI [8.75, 10.60]) and Effort (9.11, 95% CI [8.22, 10.02]) dimensions (all *p* < 0.05).

### 3.2. B.VAS Self-Report Ratings

We found no BLOCK × CONDITION interaction effect on VAS scores. However, the main effect of CONDITION (η_P_^2^ = 0.10, F(1, 211) = 22.17, *p* < 0.001) and BLOCK (η_P_^2^ = 0.33, F(1, 211) = 101.93, *p* < 0.001) emerged. VAS mental fatigue scores were higher during 100% MVC_MI_ (6.19, 95% CI [5.70, 6.70]) compared to 20% MVC_MI_ (5.24, 95% CI [4.74, 5.75]), irrespective of the block (Figure 2, *p* < 0.001). Mental fatigue VAS scores further exhibited a loglinear increase (+1.82, 95% CI [1.47, 2.17]) across blocks (*p* < 0.001).

### 3.3. Motor Performance Analysis

#### 3.3.1. Force Performances

The BLOCK × CONDITION interaction effect on peak force fell short from the statistical significance threshold (η_P_^2^ = 0.03, F(1, 138) = 4.12, *p* = 0.10). This trend originated from a marginally greater difference in peak force from the Pretest to BLOCK 2–3 during 100% MVC_MI_ compared to 20% MVC_MI_ (+11.19%, 95% CI [0.55, 23.02]). The main effect of CONDITION, however, influenced peak force values (η_P_^2^ = 0.03, F(1, 138) = 4.12, *p* = 0.04). MVC performance during 100% MVC_MI_ (2.36%, 95% CI [−1.30, 6.03]) outperformed those during 20% MVC_MI_ (−2.03%, [−4.75, 0.70]). Regarding total force, however, we found a BLOCK × CONDITION interaction (η_P_^2^ = 0.03, F(2, 211) = 3.65, *p* = 0.03), but also a main CONDITION effect (η_P_^2^ = 0.04, F(1, 211) = 8.26, *p* = 0.004). As shown in Figure 3a, during 100% MVC_MI_ the difference in total force from the Pretest (0.01%, 95% CI [−2.08, 2.08]) to BLOCK 2–3 (6.68%, 95% CI [1.55, 14.92]) was greater than the corresponding difference under the 20% MVC_MI_ (−7.11, 95% CI [−16.76, 2.53]; *p* = 0.02). There was no difference between the other post-hoc comparisons (both *p* > 0.20, Figure 3a). Total force was also superior during 100% MVC_MI_ compared to 20% MVC_MI_ (+6.86%, 95% CI [2.92, 10.80]).

#### 3.3.2. Electromyograophic Activity

We found no BLOCK × CONDITION interaction for *EMG_RATIO_*. *EMG_RATIO_* was only affected by the main effects of TEST (η_P_^2^ = 0.06, F(2, 211) = 6.11, *p* = 0.02) and CONDITION (η_P_^2^ = 0.02, F(1, 211) = 4.12, *p* = 0.03). As depicted in Figure 3b, post-hoc contrasts revealed reduced EMG_RATIO_ during the Pretest (0.00%, 95% CI [−2.84, 2.84]) compared to BLOCK 2–3 (14.00%, 95%CI [1.22, 26.80]) and BLOCK 4 (12.00%, [4.55, 19.44]) (both *p* = 0.009), and higher EMG_RATIO_ during 100% MVC_MI_ (11.84, 95% CI [4.22, 18.55]) compared to 20% MVC_MI_ (3.82, 95% CI [−0.14, 7.77]; *p* = 0.01).

### 3.4. Motor Imagery Ability

#### 3.4.1. Subjective Measures

MI ease ratings were influenced by the BLOCK × CONDITION interaction (η_P_^2^ = 0.02, F(1, 210) = 4.45, *p* = 0.03). Post-hoc analyses revealed that BLOCK negatively predicted MI difficulty ratings during 100% MVC_MI_ (−0.45, 95% CI [−0.96, 0.05]), whereas BLOCK positively predicted MI difficulty ratings during 20% MVC_MI_ (0.23, 95% CI [−0.26, 0.72]; *p* = 0.03, Figure 4a). For ratings of MI vividness, the linear mixed effects analysis only revealed a CONDITION × VAS ratings interaction (η_P_^2^ = 0.03, F(1, 209) = 6.86, *p* = 0.009). This was due to a positive predictive relationship between VAS mental fatigue scores and MI difficulty ratings during 100% MVC_MI_ (0.11, 95% CI [0.00, 0.23]), whereas the relationship was negative under the 20% MVC_MI_ (−0.06, 95% CI [−0.18, 0.05]; Figure 4b, *p* = 0.008).

#### 3.4.2. Physiological Data

The BLOCK × CONDITION influenced HR values (η_P_^2^ = 0.02, F(1, 162) = 3.62, *p* = 0.05). The negative relationship between BLOCK and HR during 20% MVC_MI_ (−1.38, 95% CI [−2.38, −0.38]) was more pronounced than that during 100% MVC_MI_ (−0.06, 95% CI [−1.00, 0.88]) (*p* = 0.05, Figure 5a). There was no predictive influence of VAS scores on HR (*p* > 0.05).

*EDR_AMP_* were not affected by the BLOCK × CONDITION interaction, which fell short form the statistical significance threshold in spite of a comparable effect size to that recorded for HR measures (η_P_^2^ = 0.02, F(1, 109) = 2.56, *p* = 0.10). This could be due to the positive relationship between VAS scores and *EDR_AMP_* during 100% MVC_MI_ (+0.09 μS, 95% CI [0.00, 0.19]), which appeared more pronounced to that recorded during 20% MVC_MI_ (0.01 μS, 95% CI [−0.07, 0.09]; Figure 5b). There was also a main effect of BLOCK (η_P_^2^ = 0.09, F(1, 109) = 11.22, *p* = 0.001), which was positively associated with *EDR_AMP_* (+0.50 μS·bloc^−1^, 95% CI [0.14, 0.84]).

## 4. Discussion

The present study was designed to provide a quantitative investigation of the development of mental fatigue induced by a MI practice session. We sought to disentangle whether mental fatigue might negatively impact maximal isometric force performance and MI ability. We implemented two practice conditions involving MI of maximal vs. submaximal isometric contractions.

We first observed a logarithmic pattern of mental fatigue increase throughout the MI practice session. This was measured from subjective mental fatigue scores on a VAS, which represents a reliable and frequently used measure to study mental fatigue. Specifically, data indicate that most of the mental fatigue developed after 40 MI trials for both MI practice conditions. After 40 trials, mental fatigue plateaued within a range of 6–8 points out of 10. Such ceiling effect might account for the bouts of passive recovery implemented in the design, since [0, 10] s passive recovery periods were administered after each MI trial. A seminal finding is that, at the session level, mental fatigue VAS scores elicited by repeated MI practice of maximal isometric contractions were superior to that elicited by submaximal contractions. This was confirmed by the subjective scores on the fatigue subscale of the mood questionnaire, where a greater increase was found for MI of maximal isometric contractions. The idea that MI of more demanding motor tasks elicits greater levels of neurophysiological arousal is not new. Past experiments emphasized greater autonomic nervous system arousal, increased corticospinal facilitation and increased residual somatic activity during MI of more demanding motor tasks [48,49,50]. An original finding of the present study is that the amount of effort required to complete imagined actions is mirrored by mental fatigue. This confirms the embodied nature of MI and corroborates past observations of comparable mental fatigue profiles after physical practice or MI training [15]. From a practical standpoint, this suggests that dose–response relationships guidelines for MI intervention in sports and rehabilitation should account for the demands of the tasks targeted by the MI intervention. More demanding motor tasks would thus elicit greater levels of mental fatigue, even when the task is mentally imagined. This advocates for MI practice sessions of shorter durations or prolonged inter-trial rest periods, hence bringing experimental evidence in support of practice guidelines published for both healthy and clinical populations [11].

Another critical finding is that mental fatigue did not negatively impact maximal isometric force. Under both MI practice conditions, we found no decrement in maximal isometric force at the intermediate or final performance assessments. This was attested by the peak force and the total force measures over the [0, 10] s time window of each maximal isometric force trial. This result invalidates the working hypothesis that force performance evaluated using multiple trials or assessed from the total force over time windows of several seconds, would be more susceptible to the negative influence of mental fatigue. This finding, however, confirms earlier observations by Rozand et al. [12] and supports the independence between mental fatigue and neuromuscular factors underlying maximal isometric force performance. Particularly, mental fatigue did not elicit central fatigue, which refers to supraspinal and spinal mechanisms that contribute to decreased force production. Possibly, physical efforts of short durations and high intensity may be less susceptible to mental fatigue compared to the repetition of submaximal efforts in endurance paradigms [14,15]. This postulate is consistent with brain homeostasis perturbation underlying mental fatigue increases reported in endurance paradigms [51,52], but also with the postulates of psychobiological fatigue models which account for decisional and motivational factors associated with increased perceived exertion [53]. Here, we found that MI practice of maximal isometric contractions facilitated force performance compared to MI practice of submaximal contractions. MI practice of maximal contractions yielded increased peak force compared to MI practice of submaximal contractions at the session-level, while total force improvement from the first to the second performance evaluation during MI of maximal contractions outperformed that recorded during MI of submaximal contractions. There are now several reports of the priming effects of MI practice on muscle activation and maximal isometric force performance [35,36]. Former experiments provided evidence for short-term plasticity mediating short-term force improvement through MI practice, particularly increased cortical gain over motor units through the downregulation of low-threshold presynaptic inhibition of alpha motor neurons [54,55,56]. In the present study, we further observed enhanced muscle activation during MI of maximal contractions compared to submaximal contractions, while the EMG activation ratio between agonists and antagonists exhibited a similar pattern to that of the total force over the course of the successive assessments under the two MI practice conditions.

An important question relates to whether the development of mental fatigue during MI practice influenced subjective and objective indexes of MI ability. Mental fatigue appeared to negatively impact MI quality of submaximal isometric contractions, but, surprisingly, facilitated MI of maximal isometric contractions. Participants reported decreased difficulty to engage in MI and increased vividness during the MI practice condition focusing on maximal isometric contractions, whereas a reverse pattern emerged for MI of submaximal isometric contractions. MI quality improvements during MI of maximal isometric contractions is consistent with the marginally positive association between subjective ratings of mental fatigue and the amplitude of electrodermal responses (Figure 5b). This challenges the hypothesis that mental fatigue negatively impacts MI quality, spurred by the generally deleterious effects of mental fatigue onset on cognitive performances [57,58,59]. Enhanced MI quality is a plausible account to the priming effects of MI on force performance. Functional brain imaging experiment provided evidence that enhanced MI quality is mediated by more focused and intense recruitment of brain motor regions [28], while transcranial magnetic stimulation attested enhanced corticospinal facilitation in good imagers [60,61]. Enhanced MI quality could thus potentiate the priming effects of MI on force performance, particularly compared to MI of submaximal contractions. Yet, why repeated MI practice of maximal isometric contractions enhanced MI quality despite the progressive onset of mental fatigue remains unclear. Past experiments demonstrated that fatigability of upper and lower limb muscles elicited by resistance training exercises exhibited positive effects on MI ability [62]. There is indeed evidence that under circumstances of increased perceived exertion, athletes tend to reallocate their attentional resources to internal task constraints at the expense of afferent sensory signals [53,63,64]. An alternative explanation would be the reciprocal influence of improved force performance through MI priming on MI ability. From a fundamental standpoint, MI is known to reproduce internal modeling operations underlying the preparation and online control of voluntary movements [65]. The inverse model associates motor intentions to motor commands, while the forward model associates motor commands to their sensory consequences. Several experiments attest that MI ability benefits from the internal model updates provided by the administration of short bouts of physical practice [16,66]. Henceforth, improved motor performance during MI practice of maximal contractions is susceptible to improve, in turn, MI ability. By contrast, MI of low-demand physical tasks, such as isometric contractions at 20% of MVC, could result in reduced physiological arousal due to boredom, hence reducing MI quality [67]. This is consistent with heart rate decrease across blocks of MI practice, which was only observed during MI of submaximal contractions.

From a practical standpoint, how mental fatigue states should be considered when designing practice guidelines for mental training is ambivalent. Although the emergence of mental fatigue did not mitigate its facilitatory effects on force performance, most of the maximal isometric force improvements were recorded between the first and the second blocks of MI practice, where mental fatigue was still low. A main limitation of the present design is that we indexed mental fatigue from subjective indexes only, although autonomic nervous system recordings could also be used as objective indexes of mental fatigue [68]. Yet, increased phasic skin conductance may not have indexed the mental fatigue, since the association with increased mental fatigue scores was only measured during MI of maximal isometric contractions. Consideration for additional objective indexes of mental fatigue elicited by MI practice could be a necessary step to better understand the neural interplay between the two processes. For instance, implementing electroencephalography measures of ongoing brain oscillations might contribute to further elucidating the influence of mental fatigue on MI ability and motor performances [69,70].

## 5. Conclusions

Present findings demonstrated the critical influence of the MI content on the development of mental fatigue and its effects on MI ability and force performance. We found facilitatory effects of MI focusing on intense physical demands compared to low physical demands, in spite of a loglinear emergence of mental fatigue under both conditions. An important research avenue is the longer-term examination of the influence of mental fatigue elicited by MI practice on the outcome of MI interventions. This particularly applies to MI interventions in neurorehabilitation settings. Indeed, susceptibility to mental fatigue could be exacerbated, for instance in stroke patients, where MI has gained consideration as an adjunct intervention to conventional rehabilitation protocols.

## Figures and Tables

**Figure 1 brainsci-13-00996-f001:**
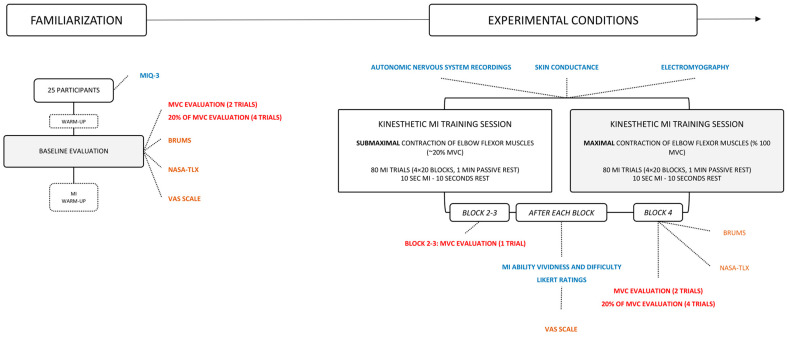
Flowchart of the experimental design.

**Figure 2 brainsci-13-00996-f002:**
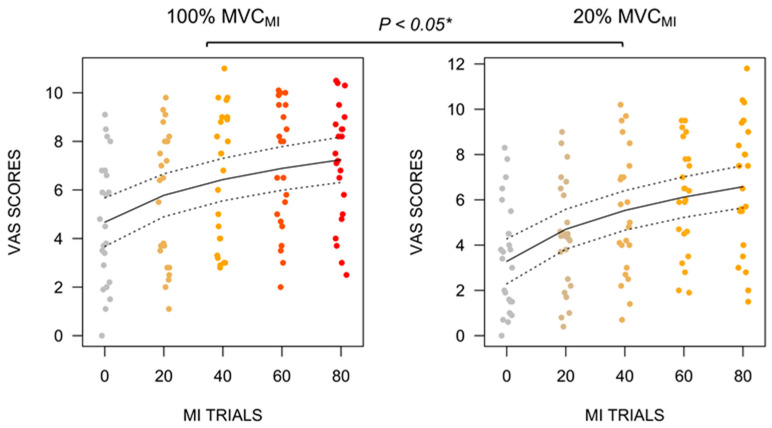
Stripchart depicting the loglinear increase in mental fatigue VAS scores from the 1st to the 4th block of the MI practice session for both experimental conditions. The increase was more pronounced during 100% MVC_MI_ compared to 20% MVC_MI_, particularly over the course of the first 3 blocks of MI practice. VAS = Visual Analogue Scale. MI = Motor Imagery. Regression slopes are shown with 95% confidence intervals (dotted lines). * *p* < 0.05.

**Figure 3 brainsci-13-00996-f003:**
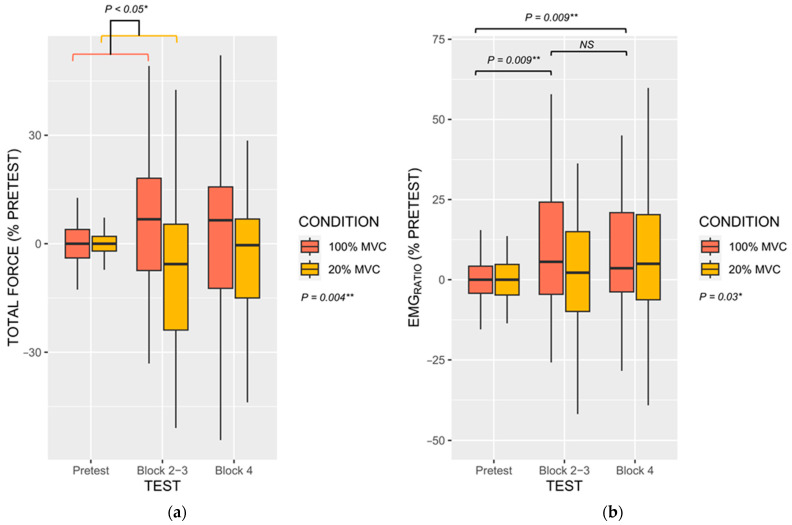
(**a**) Total force values expressed as percentage of the Pretest maximal isometric trials recorded during performance assessments throughout the MI practice session, for both experimental conditions. (**b**) *EMG_RATIO_* between agonist and antagonist activation, expressed as percentage of the Pretest maximal isometric trials, recorded during performance assessments throughout MI practice session, for both experimental conditions. Both variables displayed a very comparable pattern of change across assessments, i.e., greater improvement from the pretest to the MVC trial performed between blocks 2 and 3 during 100% MVC_MI_ compared to 20% MVC_MI_. Improvements appeared maintained from the MVC trial performed between blocks 2 and 3 and MVC trials performed after block 4. * *p* < 0.05, ** *p* < 0.01, NS = Not statistically significant.

**Figure 4 brainsci-13-00996-f004:**
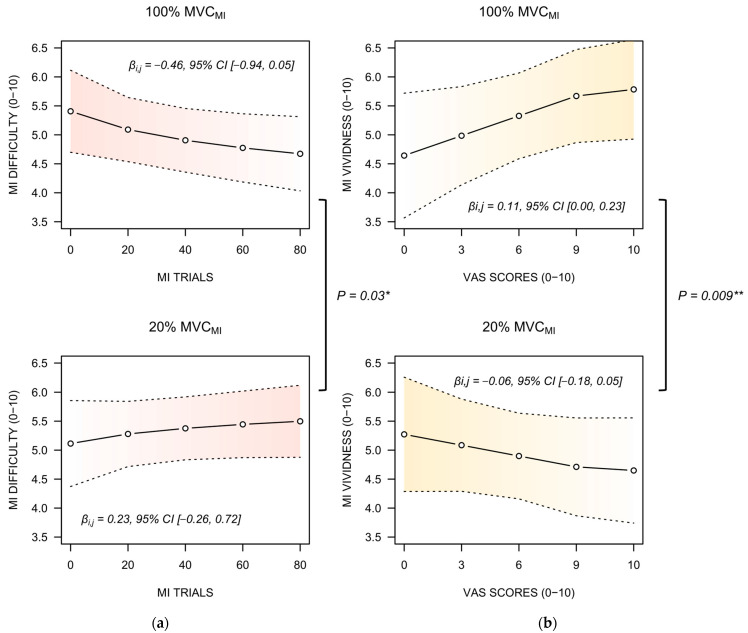
(**a**) Regression slope describing the relationship between MI trials blocs and the perceived difficulty of MI under 100% MVC_MI_ and 20% MVC_MI_. Regression slopes are shown with 95% confidence intervals (dotted lines). (**b**) Regression slope describing the relationship between blocs of MI trials and the perceived MI vividness under 100% MVC_MI_ and 20% MVC_MI_. A reverse pattern of change emerged, for both subjective indexes of MI ability, across the successive blocks of MI practice between 100% MVC_MI_ and 20% MVC_MI_. * *p* < 0.05, ** *p* < 0.01.

**Figure 5 brainsci-13-00996-f005:**
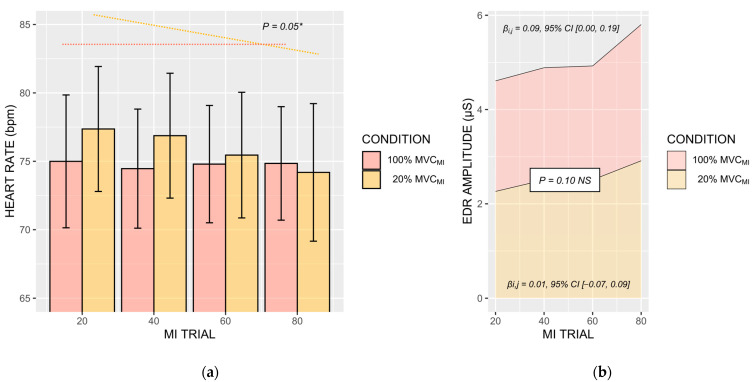
(**a**). Barplot of mean heart rate values recorded during MI trials, under 100% MVC_MI_ and 20% MVC_MI_ conditions (**b**) Electrodermal amplitudes during MI trials, under 100% MVC_MI_ and 20% MVC_MI_ conditions. A decrease in HR across the successive blocks of the MI practice session was found only during 20% MVC_MI_. * *p* < 0.05.

## Data Availability

The data associated with the paper are not publicly available but remain available from the corresponding author on reasonable request.

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
