# Peer review of "A Quantitative Investigation of Mental Fatigue Elicited during Motor Imagery Practice: Selective Effects on Maximal Force Performance and Imagery Ability"

_brainsci, 2023, doi:10.3390/brainsci13070996_

Round 1
Reviewer 1 Report
As a first point, there is no doubt that this research is extremely valuable when it comes to understanding the causes of mental fatigue.
To help in the process of providing a better understanding of this topic, it is believed that some points should be made:
1) It would be useful to include in the abstract the number of participants who participated in the study
2) It would be a good idea to add "mental fatigue" as one of your keywords.
3) Line 55, Nakashima is mentioned and shared. As a result of mental fatigue, a significant reduction in response times has been observed at the neuromuscular level which, evidently, results in a decrease in subsequent actions as well. It appears that this concept has been demonstrated recently here https://doi.org/10.3390/ijerph192114360.
4) On line 101, it is unclear if the 24 healthy adults had reported poor quality or quantity of sleep in the previous periods. Some studies (e.g. 10.3389/fnbeh.2022.945661) have suggested that this relationship may be significant.
The authors are given the opportunity to decide if these evaluations are likely to be helpful in improving the work in the future and if so, how.
Author Response
REVIEWER#1
As a first point, there is no doubt that this research is extremely valuable when it comes to understanding the causes of mental fatigue. To help in the process of providing a better understanding of this topic, it is believed that some points should be made.
We are thankful to Reviewer 1 for the positive evaluation of the original version of the manuscript. We agree with the suggestions provided in his/her review, all of which brought changes into the revised version of the manuscript. Please see below a point-by-point answer.
1) It would be useful to include in the abstract the number of participants who participated in the study
Information related to the number of participants has been included in the abstract of the revised version of the manuscript (p. 1).
2) It would be a good idea to add "mental fatigue" as one of your keywords.
“Mental fatigue” was included in the keywords of the revised manuscript.
3) Line 55, Nakashima is mentioned and shared. As a result of mental fatigue, a significant reduction in response times has been observed at the neuromuscular level which, evidently, results in a decrease in subsequent actions as well. It appears that this concept has been demonstrated recently here https://doi.org/10.3390/ijerph192114360.
We are thankful to Reviewer 1 for underlining this work, which indeed represents a relevant addition to the literature cited in the original version of the manuscript. The reference by Miggliaccio et al. (2022) is now incorporated in the revised manuscript (p. 2).
4) On line 101, it is unclear if the 24 healthy adults had reported poor quality or quantity of sleep in the previous periods. Some studies (e.g. 10.3389/fnbeh.2022.945661) have suggested that this relationship may be significant.
We admit that we did not control sleep quality using standardized questionnaires, such as the Pittsburg Sleep Quality Index (Buysse t al., 1989). However, we can attest that no participants in the present study was sleep deprived based on the subscale of the BRUMS questionnaire. Indeed, scores on item number 25 of the BRUMS, i.e. “sleepy”, accounting for the CONFUSION subscale, revealed very low levels of sleepiness in participants (0.59 ± 0.90 on the 4-points scale). This low level of sleepiness remained constant across experimental sessions (20% MVCMI: 0.47 ± 0.75; 100% MVCMI: 0.71 ± 1.03). While we did not include this data in the original version of the manuscript, since we focused on subscales scores (instead of by-item scores) of the BRUMS, we considered this data in the revised manuscript.
In the results section of the revised manuscript, we thus included data related to item 25 of the BRUMS to provide a control measure attesting the absence of acute sleep deprivation during both experimental conditions (p. 6). The reference suggested by Reviewer 1 was also included in the revised manuscript (p. 3).
Reviewer 2 Report
The authors aimed at investigating mental fatigue during motor imagery (MI) of isometric force contractions. Participants performed 20% MVCMI and 100% MVCMI trials. Mental fatigue increased logarithmically, more so during 100% MVCMI. Maximal force improved in 100% MVCMI and remained unchanged in 20% MVCMI. MI ability improved in 100% MVCMI but decreased in 20% MVCMI. They concluded that mental fatigue didn't hinder MI's benefits on maximal force or high-demand tasks but impaired MI vividness on low-demand tasks. They designed a nice experiment, and the outcomes are also interesting. However, I have some comments to clarify some details and raise some doubts.
Comments:
Major:
- I suggest adding KMI (kinesthetic motor imagery) in both the title and abstract.
- The sample size should be calculated a priori. I should add that since the number of sessions in the sample size calculation should be considered 3 as you analyzed the data. Therefore, the sample size is small and should be increased accordingly.
- The results section starts with reporting the effects of CONDITION, TEST, and DIMENSION, which were not introduced before in the materials and methods. Please introduce all the indexes and variables that you want to analyze in the materials and methods.
- Please use the same terminology in all parts of the text. For instance: CONDITION and MI CONDITION, BLOCK, and TEST. It is confusing and makes the paper very hard to understand.
- Figure 3: TOTAL FORCE (% PRETEST) and EMGRATIO (%PRETEST) should be introduced in the materials and methods. Are they z-scores or normalized to PRESET? It is confusing.
- Why were the numbers of actual MVC trials different in blocks 2 (2 trials), 3 (2 trials), and 4 (4 trials)? You could consider the same number of trials. Even now, you can consider just the first two trials in Block4 and report the new results. Applying this, maybe you can find a significant difference between Block4 and Pretest as well.
- Why did you decide to report sometimes four different blocks and sometimes combine blocks 2 and 3 together? Is there any rationale for that?
- In the abstract, you claimed that "unexpectedly, maximal force improved during 100% MVCMI but remained unchanged during 20% MVCMI," which is not true because there is no difference between Block4 and Pretest. Do you clarify it?
Minor:
- Line 11: MI should be introduced.
- There are some grammatical errors that should be revised.
- The quality of fig. 1 is not good. The font size is too small. Also, the protocol is not so clearly described by the figure. I suggest authors modifying fig. 1.
- Figure 2: top right title is 50% or 20%?
- In the VAS scores, how do you interpret the small difference between 100% and 20% MVCMI?
There are some grammatical errors that should be revised.
Author Response
REVIEWER#2
The authors aimed at investigating mental fatigue during motor imagery (MI) of isometric force contractions. Participants performed 20% MVCMI and 100% MVCMI trials. Mental fatigue increased logarithmically, more so during 100% MVCMI. Maximal force improved in 100% MVCMI and remained unchanged in 20% MVCMI. MI ability improved in 100% MVCMI but decreased in 20% MVCMI. They concluded that mental fatigue didn't hinder MI's benefits on maximal force or high-demand tasks but impaired MI vividness on low-demand tasks. They designed a nice experiment, and the outcomes are also interesting. However, I have some comments to clarify some details and raise some doubts.
We gratefully acknowledge Reviewer 2 for the encouraging feedback regarding the design and potential interest of the study. We took into consideration his/her comments to improve the original manuscript. All remarks yielded to substantial changes in the revised version. Please see the point-by-point answer below.
Comments:
Major:
- I suggest adding KMI (kinesthetic motor imagery) in both the title and abstract.
We acknowledge Reviewer 2’s remark. It is indeed true that we only administered kinesthetic MI, which is usually considered a more demanding and embodied form of MI (https://doi.org/10.1016/S0166-4115(08)62386-9; https://doi.org/10.1016/j.lfs.2017.04.003). Kinesthetic MI was thus deemed more susceptible to elicit mental fatigue.
We included kinesthetic MI in the abstract of the revised manuscript. However, we had concerns making the title too cumbersome by incorporating the MI modality, which we did not manipulate as a dependent variable of interest. Unless Reviewer 2 deems it mandatory, we shall like to keep the title of the manuscript identical between the original and revised version.
- The sample size should be calculated a priori. I should add that since the number of sessions in the sample size calculation should be considered as you analyzed the data. Therefore, the sample size is small and should be increased accordingly.
Reviewer 2 is right that power calculations are required to determine the sample size. We did run such a priori power calculations to determine the sample size, although these were not reported in the original version of the manuscript. More specifically, the sample size was determined to be able to detect medium to low effect sizes (i.e. 5-10% of explained variation) for the CONDITION * BLOC interaction effect on VAS scores with a power of p1-beta = 0.8 (PAGE). We used the pwr package (Champely, 2020), implemented in R, and the ad hoc functions for linear mixed effects models. This analysis yielded a sample size of 21 participants, which we increased to 24 participants in anticipation for potential dropouts. Why ~20 participants was a sufficient sample size to achieve 80% statistical power might account for the fact that we implemented a counterbalanced design. Indeed, within-group comparisons require reduced sample sizes compared to between-group comparisons to achieve a comparable statistical power for a given effect size. This was underlined in psychology experiments comparing within- and between-subjects designs (Campbell et al., 2004). Between-subjects designs typically require 4 to 8 times more participants than within-subjects designs (Bellemare, Bissonnette & Kröger, 2014).
Information related to power considerations was included in the statistical subsection of the methods of the revised manuscript (p. 6).
- The results section starts with reporting the effects of CONDITION, TEST, and DIMENSION, which were not introduced before in the materials and methods. Please introduce all the indexes and variables that you want to analyze in the materials and methods. Please use the same terminology in all parts of the text. For instance: CONDITION and MI CONDITION, BLOCK, and TEST. It is confusing and makes the paper very hard to understand.
Reviewer 2 is correct regarding inconsistencies in the names of the factors between the methods and results section of the original manuscript. Information was lacking regarding the description of the main effect of DIMENSION used in the analysis of scores obtained from the standardized questionnaires. The DIMENSION factor was included to control for the effect of subscales in BRUMS and NASA-TLX questionnaires (p. 6).
In the revised manuscript, the main and interaction effects for the CONDITION, TEST and DIMENSION factors are introduced in the methods (i.e. statistical analysis section, p. 6). All dependent variables are introduced in the methods and reported in dedicated sections of the results. The results section of the revised manuscript no longer contains factor names which are not described previously in the methods.
- Figure 3: TOTAL FORCE (% PRETEST) and EMGRATIO (%PRETEST) should be introduced in the materials and methods. Are they z-scores or normalized to PRESET? It is confusing.
Reviewer 2 is right that this important information was missing from the original manuscript. The total force and EMGRATIO were indeed normalized relative to the Pretests. Clarification statements were brought to the methods of the revised manuscript to better introduce these variables and the normalization process.
- Why were the numbers of actual MVC trials different in blocks 2 (2 trials), 3 (2 trials), and 4 (4 trials)? You could consider the same number of trials. Even now, you can consider just the first two trials in Block4 and report the new results. Applying this, maybe you can find a significant difference between Block4 and Pretest as well.
There was actually an error in the description of the number of trials administered during the motor performance evaluations in the methods section of the original manuscript. The number of trials was as follows: 2 trials for the assessment before the MI practice session, 1 trial for the assessment between Blocks 2 and 3, and 2 trials after Block 4 (p. 4). Limiting the number of force performance trials administered between Blocks 2 and 3 was intentionally designed to limit the amount of time spent between blocks in the absence of MI practice, thus limiting the potential recovery from mental fatigue induced by repeated MI practice at the session-level (p. 4).
Clarifications regarding the number of trials and the rationale for limiting the number of trial for motor performance evaluations between Blocks 2-3 was brought to the methods of the revised manuscript (p. 4).
- Why did you decide to report sometimes four different blocks and sometimes combine blocks 2 and 3 together? Is there any rationale for that?
Motor performance evaluations referring to force performance reported as “Block 2-3” actually refers to measures occurring between these two blocks. There was never grouping of any kind of measures between blocks.
We paid specific attention to clarify information related to measurement points in the methods of the revised manuscript (p. 4). We believe that addressing Reviewer’s preceding comment should hopefully also help to address this query in the revised manuscript.
- In the abstract, you claimed that "unexpectedly, maximal force improved during 100% MVCMI but remained unchanged during 20% MVCMI," which is not true because there is no difference between Block4 and Pretest. Do you clarify it?
Reviewer 2 is right that the BLOCK*CONDITION interaction effect on the total force reported in the results section of the original manuscript originates from greater improvements under 100% MVCMI than 20% MVCMI from the Pretest to Block 2-3 (p. 8). Noteworthy, we applied planned post hoc comparisons and accounted for the order of motor performance evaluations (i.e. Pretest trials vs. Block 2-3 trial, and Block 2-3 trial vs. Block 4 trials). As shown in Figure 3, progress from the Pretest to Block 2-3 is largely maintained in Block 4 during 100% MVCMI. By contrast, force performance (total force) remained unchanged throughout the MI practice session during 20% MVCMI. This accounts for the main CONDITION effect reported at the session-level in the results (p. 7).
Considering the pattern of force performance changes across repeated measures of the design, particularly in total force (NB. peak force exhibited a very similar pattern, yet without crossing the statistical significance threshold), underlining in the abstract that force improvements occurred under 100% MVCMI but not during 20% MVCMI appeared consistent with what the data actually show. To address Reviewer 2’s concern, we rephased our statements in the abstract of the revised manuscript, which now emphasizes that most improvements occurred between the first and second force performance assessments (p. 1).
Minor:
- Line 11: MI should be introduced.
- There are some grammatical errors that should be revised.
- The quality of fig. 1 is not good. The font size is too small. Also, the protocol is not so clearly described by the figure. I suggest authors modifying fig. 1.
- Figure 2: top right title is 50% or 20%?
- In the VAS scores, how do you interpret the small difference between 100% and 20% MVCMI?
Comments on the Quality of English Language
There are some grammatical errors that should be revised.
We thank Reviewer 2 for underlining these typos. English was amended by a native speaker, and we corrected the title for 20% in Figure 2. The quality of all figures was upgraded to 500 ppi. Differences in VAS scores are discussed in the discussion section, but an additional statement related to interpretation aspects was incorporated in Figure 3 caption.
Reviewer 3 Report
The authors present the article entitled “A quantitative investigation of mental fatigue elicited during motor imagery practice: selective effects on maximal force performance and imagery ability”
This study examines the development of mental fatigue during MI of isometric force contractions performed with the dominant upper limb.
The article presents the following concerns:
-
The text must be written in the third person or passive voice.
-
Add hyperlinks to tables, figures, and references.
-
At the end of the introduction describe the structure of the text.
-
It is necessary to reduce the percentage of duplication to less than 20% since it currently has 39%, according to Turnitin.
-
In summary, do not use abbreviations or define them, such as MI and MVCMI.
-
Add a short introduction between sections.
-
Please reduce the length of the paragraphs, as such long sections make it challenging to read.
-
Please clarify and highlight the controversy or problem this research tries to highlight in the introduction section.
-
Authors are recommended to place in section 1 the most important or outstanding research findings.
-
The quality of the figures must be improved.
-
EEG acquisition is not described in section 2.3.2.1. Please add the missing information.
-
The sentence “...MI represents an energy intensive cognitive process…“ can be justified with the following reference: A new approach for motor imagery classification based on sorted blind source separation, continuous wavelet transform, and convolutional neural network.
-
Please, introduce the emg signals before the line 200 for example, it can be used the following fresh emg references: A study of computing zero crossing methods and an improved proposal for emg signals; A novel methodology for classifying emg movements based on svm and genetic algorithms; A study of movement classification of the lower limb based on up to 4-emg channels.
-
It needs to be clarified how MI is measured or by what means this measurement is made. The authors are asked to clarify this point.
-
It is necessary to carry out a description and analysis of the displayed graphs.
-
Please review section 4, as there is a combination of results analysis and research discussion.
-
line 38: The abbreviation”i.e.” seems to be incorrectly punctuated. Consider changing the punctuation.
-
line 45: Even a knowledgeable audience may be unfamiliar with the word “deleterius” you should be use “harmful”
-
line 52: The use of “and/or” is severely frowned upon in formal writing. Consider using only one conjunction or rewriting the sentence.
-
line 462: The phrase “is likely influencing” may be wordy. Consider changing the wording by “likely influences”
Author Response
REVIEWER#3
Comments and Suggestions for Authors
The authors present the article entitled “A quantitative investigation of mental fatigue elicited during motor imagery practice: selective effects on maximal force performance and imagery ability”. This study examines the development of mental fatigue during MI of isometric force contractions performed with the dominant upper limb.
The article presents the following concerns:
- The text must be written in the third person or passive voice.
- Add hyperlinks to tables, figures, and references.
- At the end of the introduction describe the structure of the text.
Reviewer 3 is right. The revised manuscript was written in the third or passive voice. We are not certain that hyperlinks to tables and Figures are required for draft manuscripts. Indeed, unless we are mistaken, hyperlinks are usually added after acceptance during the final stages of the publication process. Also, we are not certain what Reviewer 3 implies by “describe the structure of the text” at the end of the introduction. If further request by Reviewer 3 is formulated, we remain open to incorporate further changes in the introduction.
It is necessary to reduce the percentage of duplication to less than 20% since it currently has 39%, according to Turnitin.
We acknowledge Reviewer 3’s standpoint. We admit that we do not have access to duplication metrics provided by software such as Turnitin. The present manuscript was written without any form of plagiarism. While it is possible that some degree of overlap exists between some sections of the present manuscript and some of our former studies (e.g. statistical analysis section), this does not applies to discursive sections such as the introduction and discussion. Redundancies in the writing styles might only reflect common methodologies between the present work and past research by our group.
We do not know of the specificities of Turnitin, yet if Reviewer 3 provides specific sections which could raise concern with regards to potential plagiarism, we would happily address them for we are totally convinced that we did not duplicate any statement from other research groups.
- In summary, do not use abbreviations or define them, such as MI and MVCMI.
- Add a short introduction between sections.
- Please reduce the length of the paragraphs, as such long sections make it challenging to read.
- Please clarify and highlight the controversy or problem this research tries to highlight in the introduction section.
- Authors are recommended to place in section 1 the most important or outstanding research findings.
We thank Reviewer 3 for the careful reading and instructions regarding the write-up of the manuscript. We first checked that all acronyms are previously defined before being employed in the revised version, particularly “MI” and “MVC”. Second, we agree that limiting paragraph’s length is required for cumbersome or non-essential information. Reading through the revised manuscript that incorporated comments and requests from other Reviewers, we did not find which specific section or parts were redundant or superfluous. We have concerns that deleting statements or cutting through paragraphs would make the manuscript less comprehensive. Nevertheless, we would gladly consider any specific request by Reviewer 3 to work on sections that would remain challenging to read. Eventually, we emphasized the controversy or problem the present study addresses in the introduction of the revised manuscript (p. 2), where the most important research findings that relate to the present work are cited. Regarding this last point, it is noteworthy that we incorporated the suggestions by Reviewers 1 and 2 regarding the state of the art, which should also hopefully address Reviewer 3’s request.
The quality of the figures must be improved.
Reviewer 3 is right, for this point which was also underlined by Reviewer 2. The resolution of the figures has been improved to 500 ppi in the revised manuscript.
EEG acquisition is not described in section 2.3.2.1. Please add the missing information.
There was actually no EEG acquisition, we made a typo between “electroencephalographic” and “electrocardiographic” in the title of section 2.3.2.1. This is now addressed in the revised manuscript (p. 5).
The sentence “...MI represents an energy intensive cognitive process…“ can be justified with the following reference: A new approach for motor imagery classification based on sorted blind source separation, continuous wavelet transform, and convolutional neural network.
We agree on the relevance of the reference, which was included in the revised manuscript (p. 1).
Please, introduce the emg signals before the line 200 for example, it can be used the following fresh emg references: A study of computing zero crossing methods and an improved proposal for emg signals; A novel methodology for classifying emg movements based on svm and genetic algorithms; A study of movement classification of the lower limb based on up to 4-emg channels.
In most research manuscripts, description of the acquisition procedures for neurophysiological signals is usually provided in the methods section. Furthermore, we feel uncomfortable incorporating references to EMG methods that we did not use nor have expertise in. While we did not modify this point in the revised manuscript, we acknowledge Reviewer 3 for underlining interesting research in the field of EMG signals classification.
It needs to be clarified how MI is measured or by what means this measurement is made. The authors are asked to clarify this point.
We agree with Reviewer 3. Methods related to MI ability assessments were provided in the original manuscript. We first provided a general framework for MI ability assessments in the introduction (p. 2). Second, the significance of Likert-type scales used for MI vividness and ease/difficulty measures is provided in the methods (p. 4). Third, the significance of EDR as objective marker of MI ability is mentioned explicitly in the methods (p. 5).
We paid specific attention that any relevant detail necessary to reproduce MI ability assessment procedures implemented in the present design appeared in the methods of the revised manuscript. Yet, we remain open and willing to incorporate additional information deemed necessary by Reviewer 3 to clarify how MI ability was measured.
It is necessary to carry out a description and analysis of the displayed graphs.
We agree that Figure captions were too succinct. Additional details were incorporated in the figures, although it was difficult to establish balance between statements related to the results and research discussion.
Please review section 4, as there is a combination of results analysis and research discussion.
We reviewed the results section and amended any statement that could relate to interpretation of the data rather than description of the results.
Comments on the Quality of English Language
- line 38: The abbreviation”i.e.” seems to be incorrectly punctuated. Consider changing the punctuation.
- line 45: Even a knowledgeable audience may be unfamiliar with the word “deleterius” you should be use “harmful”
- line 52: The use of “and/or” is severely frowned upon in formal writing. Consider using only one conjunction or rewriting the sentence.
- line 462: The phrase “is likely influencing” may be wordy. Consider changing the wording by “likely influences”
Thank you for underlining these typos. All were amended in the revised version of the manuscript. Regarding “i.e.”, it seems that it can be employed without additional punctuation but please do not hesitate to suggest any phrasing improvements. Also, we believe that most academic researchers might be familiar with the term “deleterious”. Yet, we considered Reviewer 3’s recommendation and employed “negative” to avoid repetition. We agree with the use of “and/or”, which was rephrased in the revised manuscript. The only instance of “and/or” in the revised manuscript is for the Disclaimer/Publisher’s Note, which is a mandatory statement of the journal.
Round 2
Reviewer 2 Report
I am pleased to accept the revised version of the manuscript, as the authors have effectively addressed my concerns.
Reviewer 3 Report
The manuscript can be accepted for publication